# Kaempferol Enhances Sperm Post-Thaw Survival by Its Cryoprotective and Antioxidant Behavior

Štefan Baňas [1], Filip Benko [1], Michal Ďuračka [2], Norbert Lukáč [3] and Eva Tvrdá [1,*]

1 Institute of Biotechnology, Faculty of Biotechnology and Food Sciences, Slovak University of Agriculture in Nitra, Tr. A. Hlinku 2, 949 76 Nitra, Slovakia; xbanas@uniag.sk (Š.B.); filip.benko@uniag.sk (F.B.)

2 AgroBioTech Research Centre, Slovak University of Agriculture in Nitra, Tr. A. Hlinku 2, 949 76 Nitra, Slovakia; michal.duracka@uniag.sk

3 Institute of Applied Biology, Slovak University of Agriculture in Nitra, Tr. A. Hlinku 2, 949 76 Nitra, Slovakia; norbert.lukac@uniag.sk

* Correspondence: eva.tvrda@uniag.sk; Tel.: +421-37-641-4918

**Abstract:** This study examined the effects of three selected doses of kaempferol (KAE; 12.5, 25 or 50 μM) on bovine sperm motility and oxidative profile directly related to cold storage. We also elucidated the effect of KAE on the expression profiles of heat shock proteins (HSPs) 70 and 90 as well as the pro-apoptotic BCL2-associated X (BAX) protein and the anti-apoptotic B-cell lymphoma 2 (Bcl-2) protein. Frozen samples supplemented with KAE were compared with a native control (fresh spermatozoa) and a cryopreserved control, frozen in the absence of KAE. Our results demonstrate that the administration of all KAE doses led to a higher degree of sperm motility ($p < 0.05$) when compared with the cryopreserved control. The highest levels of protection of sperm DNA ($p < 0.05$), lipids ($p < 0.05$) and proteins ($p < 0.05$) were detected in samples exposed to 25 μM KAE when compared with samples frozen without KAE. Administration of 25 μM KAE led to a significant increase in HSP70 and HSP90 ($p < 0.05$) when compared with the unsupplemented frozen control. No significant differences were observed in the expression patterns of BAX; however, a significant up-regulation of Bcl-2 protein was observed in the frozen samples enriched with 25 μM KAE when compared with the cryopreserved control ($p < 0.05$). In summary, we may consider KAE as an effective agent in stabilizing the sperm membranes by preventing reactive oxygen species (ROS) overproduction in the mitochondria and subsequent oxidative damage to molecules critical for a proper sperm architecture and function. These protective properties of KAE may lead to higher post-thaw sperm activity and viability.

**Keywords:** kaempferol; cryopreservation; bull spermatozoa; oxidative stress; protein expression; apoptosis; heat shock proteins

## 1. Introduction

Sperm cryopreservation is a critical factor for the efficiency and success of artificial insemination, as it enables more effective breeding of genetically superior animals worldwide. Cold storage of ejaculates is also an essential step for in vitro methods in veterinary andrology and for preserving the genetic potential and biodiversity of protected, vulnerable, and valuable individuals [1]. Nevertheless, whilst being technically and logistically well-developed, the cryopreservation process still presents a possible risk of losing post-thaw sperm vitality and fertilizing ability [2]. Spermatozoa, based on their structural and morphological peculiarities, such as higher levels of cholesterol, lower ratio of unsaturated vs. saturated fatty acids in the plasma membranes, coupled with almost non-existing cytoplasm [3,4], may be significantly compromised by the freezing/thawing procedure both at structural and functional levels [2,5]. Besides intracytoplasmic ice formation and thermal and osmotic shock, the overproduction of reactive oxygen species (ROS) is a paramount

contributor to the damage [6]. Accordingly, high proportions of spermatozoa with per-oxidized membranes, dysfunctional proteins, and DNA breakage have been frequently observed following cryopreservation in humans as well as animals [2,7–11]. The accumulation of toxic by-products may subsequently lead to irreparable cell damage, activation of the apoptotic molecular machinery, and loss of viable male gametes [12].

Antioxidants are molecules that inhibit ROS formation or are in charge of detoxication, removal and/or reparation of the cellular damage caused by oxidative insults. While superoxide dismutase, glutathione peroxidase, and catalase are prime endogenous antioxidants that are paramount for a proper sperm function [13], numerous studies have now demonstrated that the use of plant biomolecules isolated from natural sources with high bioavailability and low cytotoxicity [14] may act as a promising support system to the antioxidant capacity of the semen sample that is compromised during the freeze–thaw procedure. Scientific studies show that natural biomolecules such as curcumin, kaempferol, naringenin, and punicalagin present with structural or functional attributes that may be advantageous in the prevention of cellular stress, such as protective effects against sperm membrane alterations induced by cryocapacitation or ability to reduce or reverse oxidative breakdown of biomolecules vital for sperm survival. Systematic screening of natural sources also reveals that natural bioactive molecules exhibit antimicrobial and pronounced antioxidant properties based on their ability to scavenge ROS and modulate intracellular antioxidant enzymes [15–20].

Recently, there has been increasing interest in the bioactive potential of kaempferol [21,22]. Kaempferol (KAE; 3,5,7-trihydroxy-2-(4-hydroxyphenyl)-4-H-1-benzopyran-4-one) has a low molecular weight (MW: 286.2 g/mol) and is found in broccoli, cabbage, beans, tomatoes, strawberries, grapes, and tea [23] as well as plants used in traditional medicine such as *Amburana cearensis* [24], *Aloe vera* [25], and *Cassia angustigolia* [26]. KAE is said to have certain health benefits such as antioxidant and anti-inflammatory properties [27], antitumor effects [28], or depigmenting activity [29]. This flavonoid neutralizes ROS, interacts with $\alpha$-tocopherol to slow its oxidation, and inhibits glutathione-S-transferase, UDP-glucuronosyltransferase, and NAD(P)H-oxidase. In addition, KAE has antibacterial properties [30].

Due to insufficient research studies of KAE properties in terms of its impact on male reproductive cells, we strived to assess the impact of KAE on central oxidative characteristics of frozen–thawed bovine sperm. Moreover, we studied any possible effects of KAE on the expression patterns of heat shock proteins (HSPs) 70 and 90 acting as primary indicators of sperm freezability [31] as well as the BCL2-associated X (BAX) protein and the B-cell lymphoma 2 (Bcl-2) protein, known to play essential roles in the regulation of sperm apoptosis [32].

## 2. Results

### 2.1. Motility Characteristics

CASA analysis revealed that cryopreservation had a significant negative effect on the sperm motion parameters (Table 1). The lowest motility, progressive motility and kinematic characteristics were observed in the cryopreserved control (CtrlC; $p < 0.05$ vs. the native control—CtrlN). On the other hand, significant improvements in sperm motility and progressive motility were observed in the presence of all three doses of KAE ($p < 0.05$ vs. CtrlC). With respect to secondary motion characteristics including path velocity, progressive velocity, and track speed, a significant improvement was recorded particularly in the samples cryopreserved in the presence of 25 $\mu$M KAE in comparison with CtrlC ($p < 0.05$).

**Table 1.** Motility characteristics of bovine spermatozoa in native state and cryopreserved in the absence or presence of different kaempferol (KAE) concentrations.

| Parameter | CtrlN | CtrlC | 12.5 μM KAE | 25 μM KAE | 50 μM KAE |
|---|---|---|---|---|---|
| MOT [%] | 96.17 ± 4.40 | 60.12 ± 5.83 [A] | 80.83 ± 4.16 [A,B] | 82.53 ± 3.14 [A,B] | 75.78 ± 4.65 [A,B] |
| PROG [%] | 76.09 ± 6.94 | 32.57 ± 5.35 [A] | 60.10 ± 5.98 [A,B] | 63.78 ± 7.02 [B] | 42.67 ± 5.76 [A,B] |
| VAP [μm/s] | 107.50 ± 8.44 | 70.43 ± 7.59 [A] | 86.97 ± 8.80 [A] | 92.17 ± 14.61 [B] | 74.93 ± 8.46 [A] |
| VSL [μm/s] | 91.23 ± 6.70 | 51.20 ± 6.62 [A] | 64.60 ± 8.33 [A] | 72.43 ± 9.77 [A,B] | 59.93 ± 6.49 [A] |
| VCL [μm/s] | 184.30 ± 16.85 | 153.90 ± 11.22 [A] | 169.10 ± 7.19 | 178.60 ± 21.61 [B] | 166.10 ± 8.86 |
| ALH [μm] | 7.86 ± 0.82 | 6.60 ± 0.36 [A] | 6.73 ± 0.53 [A] | 7.68 ± 0.54 [B] | 6.64 ± 0.69 [A] |
| BCF [Hz] | 40.80 ± 3.65 | 29.57 ± 2.32 [A] | 31.50 ± 2.71 [A] | 35.03 ± 3.53 [A,B] | 30.40 ± 3.06 [A] |
| STR [%] | 84.67 ± 7.70 | 72.44 ± 6.62 [A] | 80.69 ± 7.09 | 79.00 ± 6.16 [A] | 76.20 ± 7.10 [A] |
| LIN [%] | 51.00 ± 3.59 | 33.00 ± 2.94 [A] | 45.33 ± 3.70 [B] | 46.00 ± 5.65 [B] | 43.00 ± 3.82 [A,B] |

Mean ± standard deviation (S.D.). MOT—motility, PROG—progressive motility, VAP—path velocity, VSL—progressive velocity, VCL—track speed, ALH—lateral amplitude, BCF—beat frequency, STR—straightness, LIN—linearity. Significant if $p < 0.05$. [A]—vs. native control (CtrlN); [B]—vs. cryopreserved control (CtrlC). Five replicates from each bull and group were assessed.

### 2.2. Oxidative Profile

As seen in Figure 1, exposure to low temperature negatively affected the susceptibility of sperm DNA to oxidative damage, as the highest concentration of 8-hydroxy-2'-deoxyguanosine (8-OHdG) was detected in the CtrlC group ($p < 0.05$ vs. CtrlN). Notably, a significant decrease in 8-OHdG levels was observed with 25 μM KAE when compared with CtrlC ($p < 0.05$).

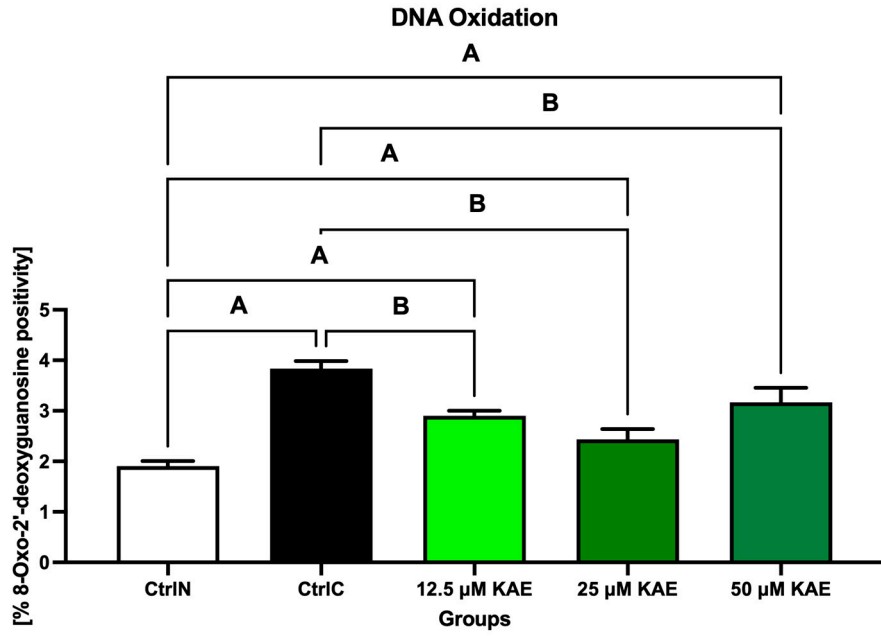

**Figure 1.** Oxidative DNA damage of bovine spermatozoa in fresh state and cryopreserved in the absence or presence of different kaempferol (KAE) concentrations. Significant if $p < 0.05$. [A]—vs. native control (CtrlN); [B]—vs. cryopreserved control (CtrlC). Each bar represents mean (±S.D.). Five replicates from each bull and group were assessed.

The highest concentration of protein carbonyls was found in the cryopreserved control (CtrlC), which differed significantly from the native control (CtrlN; $p < 0.05$; Figure 2). Although lower levels of oxidative damage to proteins were observed in all KAE-supplemented experimental groups, the lowest amount of protein carbonyls was detected after sperm exposure to 25 μM KAE, which was significant in comparison with CtrlC ($p < 0.05$).

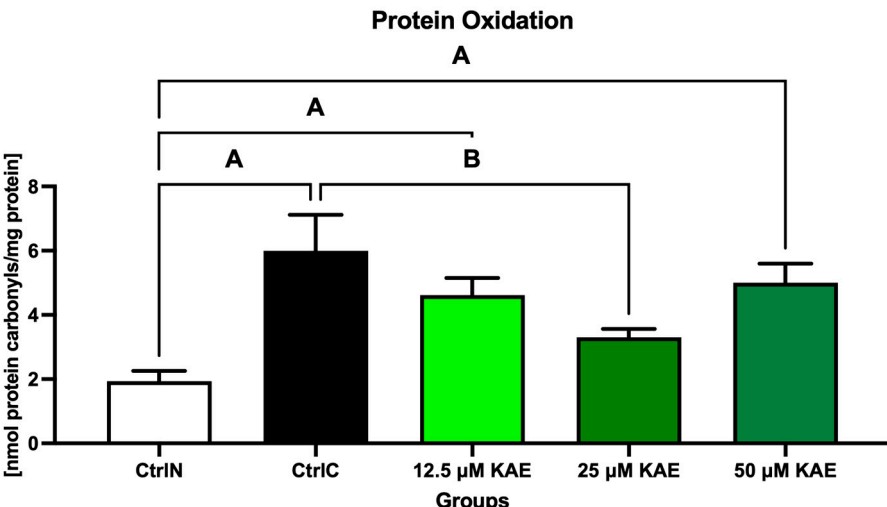

**Figure 2.** Oxidative damage to the proteins of bovine spermatozoa in fresh state and cryopreserved in the absence or presence of different kaempferol (KAE) concentrations. Significant if $p < 0.05$. [A]—vs. native control (CtrlN); [B]—vs. cryopreserved control (CtrlC). Each bar represents mean ($\pm$S.D.). Five replicates from each bull and group were assessed.

The extent of peroxidative damage to sperm lipids reflected the same trends observed in previous analyzes of the oxidative profile (Figure 3). The highest degree of oxidative impairment of lipid molecules, reflected by the highest malondialdehyde (MDA) concentration, was observed in the cryopreserved control (CtrlC) when compared with the native sperm sample (CtrlN; $p < 0.05$). Significant differences were observed between CtrlN, 12.5 μM KAE ($p < 0.05$), and 50 μM KAE ($p < 0.01$) in the comparative analysis, whereas a significant decrease in MDA was observed in the experimental group exposed to 25 μmol/L KAE in comparison to CtrlC ($p < 0.05$).

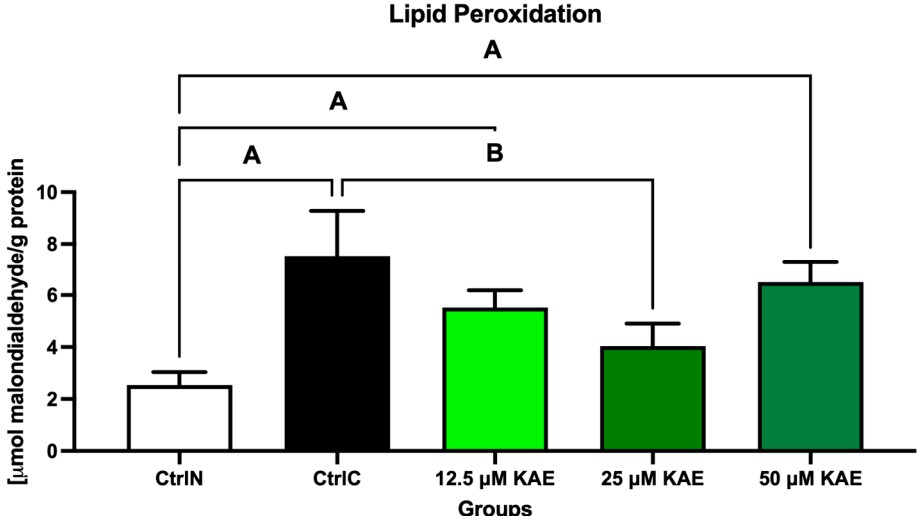

**Figure 3.** Oxidative damage to the lipids of bovine spermatozoa in fresh state and cryopreserved in the absence or presence of different kaempferol (KAE) concentrations. Significant if $p < 0.05$. [A]—vs. native control (CtrlN); [B]—vs. cryopreserved control (CtrlC). Each bar represents mean ($\pm$S.D.). Five replicates from each bull and group were assessed.

### 2.3. Protein Expression

Selected heat shock proteins as well as proteins associated with the apoptotic machinery were analyzed and semi-quantified using the Western blot technique; their expression

patterns are displayed in Figure 4. Beta actin was used as an internal control for data normalization.

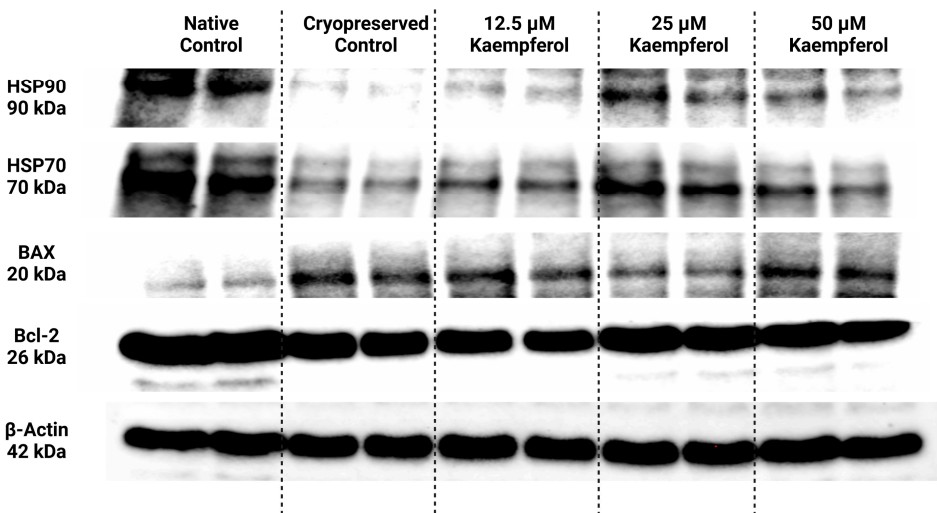

**Figure 4.** Protein expression patterns of HSP90, HSP70, BAX, Bcl-2 and β-Actin in bovine spermatozoa cryopreserved in the presence of three doses of kaempferol, as determined by Western blotting. Original photos of the gels, membranes, and blots are available as Supplementary Materials.

Data collected from Western blotting demonstrated that HSP90 expression in sperm was significantly negatively affected by low temperatures when compared to their natural state ($p < 0.05$; Figures 4 and 5). On the other hand, all doses of KAE added to the cryopreservation medium were able to protect HSPs to some extent after the freeze–thaw procedure. Notably, HSP90 protein expression was significantly increased in the experimental group containing 25 μM KAE when compared with the cryopreserved control ($p < 0.05$).

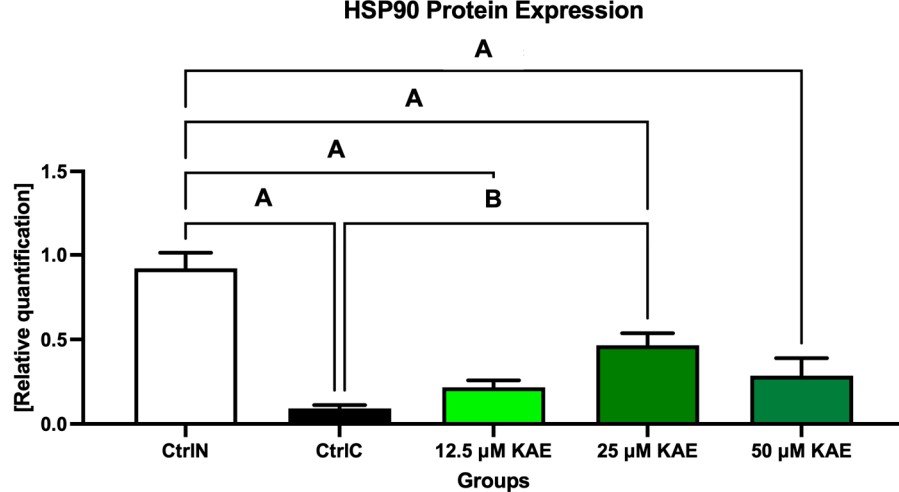

**Figure 5.** Heat shock protein 90 (HSP90) expression patterns of bovine spermatozoa in fresh state and cryopreserved in the absence or presence of different kaempferol (KAE) concentrations. Significant if $p < 0.05$. [A]—vs. native control (CtrlN); [B]—vs. cryopreserved control (CtrlC). Each bar represents mean (±S.D.). The assay was run in triplicate each with two randomly selected samples from the control and experimental groups.

A similar dose-dependent behavioral pattern was also observed for HSP70 protein expression (Figures 4 and 6). All experimental groups showed a higher expression of HSP70

compared with cryopreserved control, with the highest significance being detected in the case of 25 µM KAE in comparison with CtrlC ($p < 0.05$).

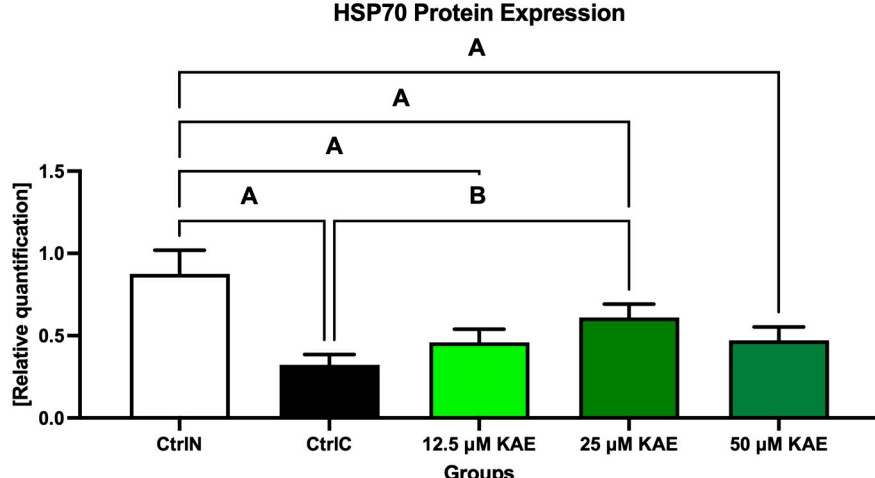

**Figure 6.** Heat shock protein 70 (HSP70) expression patterns of bovine spermatozoa in fresh state and cryopreserved in the absence or presence of different kaempferol (KAE) concentrations. Significant if $p < 0.05$. [A]—vs. native control (CtrlN); [B]—vs. cryopreserved control (CtrlC). Each bar represents mean (±S.D.). The assay was run in triplicate each with two randomly selected samples from the control and experimental groups.

As shown in Figures 4 and 7, proapoptotic BAX protein expression was increased following the cryopreservation process (CtrlC) although without statistical significance when compared to the native sperm sample (CtrlN). The addition of particularly 25 µM KAE caused the most notable although insignificant decrease in BAX expression when compared to the freeze–thawed control.

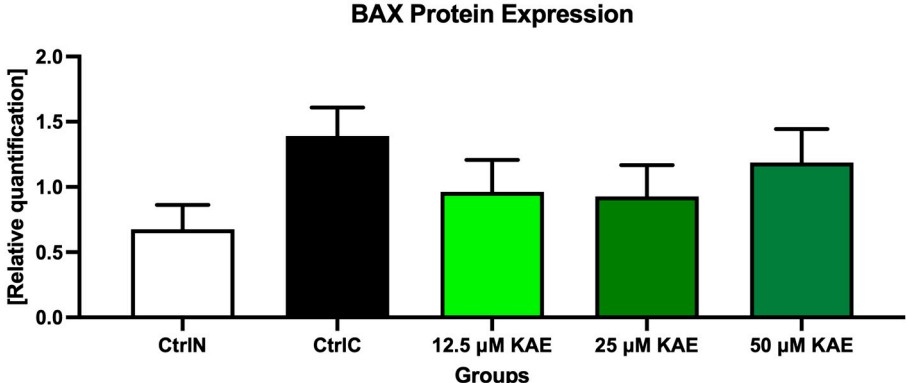

**Figure 7.** BCL2-associated X (BAX) protein expression patterns of bovine spermatozoa in fresh state and cryopreserved in the absence or presence of different kaempferol (KAE) concentrations. Each bar represents mean (±S.D.). The assay was run in triplicate each with two randomly selected samples from the control and experimental groups.

Conversely, the anti-apoptotic Bcl-2 protein was significantly underexpressed in un-treated frozen–thawed samples compared to native control ($p < 0.05$; Figures 4 and 8). A slight improvement in Bcl-2 expression pattern was observed after administration of 12.5 µM KAE and 50 µM KAE, whereas the experimental group with 25 µM KAE demonstrated a significant improvement in Bcl-2 protein levels when compared to the cryopreserved control ($p < 0.05$).

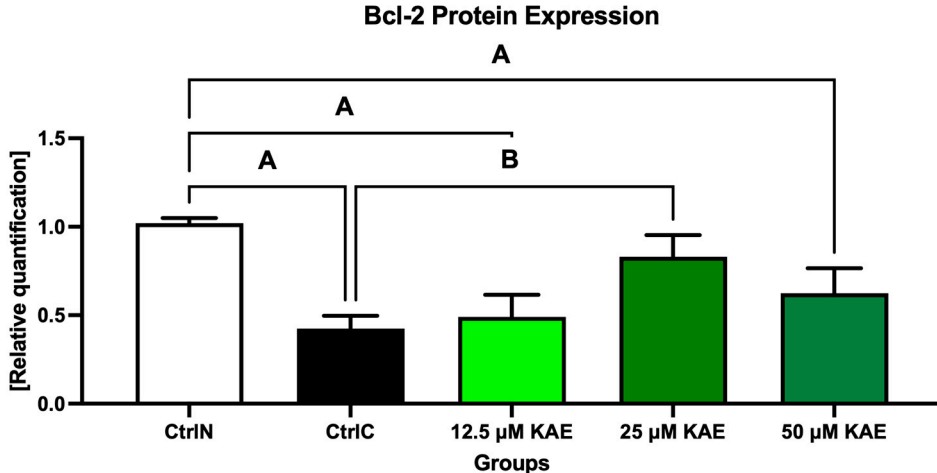

**Figure 8.** B-cell lymphoma 2 (Bcl-2) protein expression patterns of bovine spermatozoa in fresh state and cryopreserved in the absence or presence of different kaempferol (KAE) concentrations. Significant if $p < 0.05$. [A]—vs. native control (CtrlN); [B]—vs. cryopreserved control (CtrlC). Each bar represents mean ($\pm$S.D.). The assay was run in triplicate each with two randomly selected samples from the control and experimental groups.

## 3. Discussion

Our results indicate that kaempferol presents with protective roles on the sperm post-thaw survival by enhancing the motion activity and viability, especially at concentrations of 25 μM. Previous preliminary studies on this flavonoid have drawn similar conclusions. El-Raey and Azab [33] showed that concentrations of 25 and 50 μg/mL KAE significantly improved the post thaw motility, acrosomal integrity and viability index of buffalo spermatozoa. Ďuračka et al. [34] also reported beneficial effects of KAE on extended boar sperm motility in a dose-dependent manner. Moreover, a study by Jamalan et al. [35] found that the motility of human sperm exposed to aluminum chloride was restored by 10% at KAE concentrations of 25 μmol/L. This stimulating effects of KAE may be connected to its ability to modulate the mitochondrial calcium uniporter located in the sperm plasma membrane without being dependent upon ATP [36]. At the same time, it has been hypothesized that flavonoids such as KAE may increase sperm longevity by inhibiting glycolysis [37]. These mechanisms represent an alternative for natural biomolecules to stimulate mitochondrial metabolism critical for the sperm motion [36]. Nevertheless, it must be noted that bioactive flavonoid substances may act like a double-edged sword where low concentrations improve the sperm structural and functional properties, whereas high concentrations are non-effective or directly cytotoxic [35,38,39]. This evidence may, therefore, suggest that the precise effects of KAE on the sperm motility may depend on dosage, sperm processing techniques, species, or analysis time.

The collected data on the oxidative profile of spermatozoa suggest that appropriately selected KAE doses may at least partially prevent excessive oxidative insults to biomolecules essential for a proper sperm function. This is probably either because KAE may interact with ROS before they interact with sperm lipids, proteins, or DNA, or that KAE may easily become incorporated into critical sperm structures and thus offer a direct protection of vital biomolecules once ROS reach them. Accordingly, El-Raey and Azab [33] as well as Wang et al. [40] observed a strong antioxidant potential of KAE reflected by an improved total antioxidant capacity, superoxide dismutase activity and glutathione levels with a concomitant reduction in lipid peroxidation. The strong ability of KAE in preventing peroxidation of membrane lipids was also observed by Jamalan et al. [35] in spermatozoa exposed to aluminum chloride, cadmium chloride or lead chloride, resulting in increased sperm motility. Similarly, according to Ďuračka et al. [34], a rational use of KAE during semen storage may maintain a proper mitochondrial activity whilst preventing excessive

superoxide radical. Another study by Kluska et al. [41] showed that KAE derivatives from *Lens culinaris* may help protect eukaryotic cells from DNA damage by etoposide. All in all, we may agree with previously postulated hypotheses that KAE as a flavonoid may prevent oxidative damage by inhibiting and/or directly quenching excessive ROS, delay oxidation by interacting with other components of the antioxidant defense mechanisms as well as promoting the gene expression of detoxifying enzymes [33,37,40]. A significant reduction in oxidative damage to spermatozoa that persisted in the presence of KAE following the thawing procedure in this study may provide further evidence for its antioxidant properties.

Heat shock proteins (HSPs) play an essential role in reducing stress and maintaining a normal cellular activity, as HSP fluctuations directly affect the acquisition of cytoresistance [42]. Various studies have shown that HSP expression levels may be altered in cells under heat or cold stress [42–45]. The significant underexpression of both HSPs in cryopreserved controls when compared to fresh sperm in this study suggests that bovine spermatozoa may respond to the freeze–thaw process by slowly degrading HSPs present in the plasma membranes [43]. Alternatively, it has been proposed that HSPs, like other proteins, may "leak" to the extracellular milieu or semen extenders in response to cold shock. However, previous reports have not demonstrated the presence of HSP90 or HSP70 neither in the seminal plasma prior to cryopreservation, nor in the cryopreservation medium following the thawing procedure [42,45,46]. According to Zhang et al. [45,46], another reason for HSP loss may be related to the inability of frozen cells to properly carry out HSP synthesis, or the production of any proteins during freezing. As such, we may speculate that HSP70 and HSP 90 may have been consumed in order to support a proper protein assembly and/or conformation process. Furthermore, frozen- thawed semen samples contain an increased proportion of dead cells that are no longer able to carry out any protein synthesis [5]. In this study, downregulation of HSP90 and HSP70 was associated with a significant reduction in the sperm motility following thawing. This observation agrees with the hypothesis of Wang et al. [44] who suggest that a decline in HSP levels may predict low semen quality. Moreover, these data suggest that an increased pre-freezing HSP synthesis or administration may improve the sperm viability and activity following thawing [46]. Exposure of cells to cold temperature increases ROS production, impairing sperm vigor and fertility [18,47]. It has been hypothesized that HSPs may affect protein folding and movement of polypeptides across the plasma membrane [48]. Correspondingly, earlier reports suggest that HSP underexpression may lead to abnormal folding of membrane proteins and subsequent changes in membrane fluidity [49]. Since it has been hypothesized that HSPs may also play a role in the protection of antioxidant enzymes, we may speculate that reduced HSP levels in cryopreserved samples may reduce the resistance of sperm proteins, lipids, and DNA to oxidative damage caused by ROS overgeneration during the freeze–thaw process. Limited information is available with respect to the effects of natural biomolecules on the HSP expression patterns during sperm cryopreservation. In this study, KAE supplementation to the semen extender, particularly at 25 μM was shown to significantly preserve both HSPs in cryopreserved spermatozoa. Previous reports on a similar topic have revealed that anthocyanin cyanidin-3-O-β-glucopyranoside, cyanidin chloride or naringenin may exhibit antioxidant abilities, in part through an inducible expression of HSP70 in mammalian cell lines [50,51]. Further studies on biologically active molecules from natural sources such as epicatechin, quercetin or genistein may modulate HSPs [52,53]. Nevertheless, further research is necessary to elucidate their specific mechanisms of action by which they exhibit their protein-stabilizing effects.

Apoptotic machinery is thought to play a complex role in sperm cryoinjury, as the freezing and thawing procedure have been associated with the activation of specific caspases in several mammalian germ cells [10,12,54–56]. The exact apoptotic mechanisms involved in sperm cryodamage are not fully understood yet, which complicates the identification and/or removal of early apoptotic spermatozoa during assisted reproduction procedures, leading to a higher occurrence of failed fertilization and embryogenesis [55].

Interactions between the pro-apoptotic BAX and the anti-apoptotic Bcl-2 protein are considered as an important indicator of the cell fate. While the main function of Bcl-2 is to prevent the activity of pro-apoptotic proteins responsible for the formation of pores in the mitochondrial system [57], BAX functions as an apoptotic activator [58]. Our Western blot data confirm earlier studies reporting that cryopreservation supports the activation of the apoptotic machinery including the loss of mitochondrial membrane potential, activation of caspases 3, 8 and 9, alteration to the membrane permeability and phosphatidylserine externalization [59]. Interestingly, while BAX has been previously identified in bovine spermatozoa, Bcl-2 was absent [32,60]. We may associate this discrepancy with differences in the protein extraction procedure, the Western blot protocol, or the chosen antibodies. Similarly to HSPs, only a limited number of studies have been conducted to evaluate the potential of bioactive compounds to prevent or mitigate the overactivation of apoptotic machinery in cryopreserved spermatozoa. In a study by Rezaei et al. [61], 5 mmol/L L-carnitine significantly decreased BAX mRNA levels and increased Bcl-2 mRNA expression in cryopreserved mouse epididymal spermatozoa. Our previous study with epicatechin [10] revealed a significant dose-dependent improvement of the BAX-Bcl-2 ratio that was associated with a higher post-thaw sperm vitality. Finally, Redza-Dutordoir and Averill-Bates [62] suggest that lycopene being a powerful ROS scavenger prevents oxidative insults to the mitochondria, stabilizing the mitochondrial membrane potential, prevents BAX and BAD translocation, and subsequent cytochrome c release. According to our observations, KAE seems to be effective to stabilize the balance amongst the pro-apoptotic and anti-apoptotic proteins particularly at a dose oscillating around 25 μM; its higher concentrations seem to act in a contra-productive manner, increasing the levels of BAX and decreasing the Bcl-2 expression.

## 4. Materials and Methods

### 4.1. Semen Collection and Cryopreservation

Semen samples were collected from 20 healthy and sexually mature Holstein bulls (Slovak Biological Services, a.s., Nitra, Slovakia) in the winter of 2022 using artificial vagina [18]. Each sample was divided into five equal aliquots. The first, serving as the native control (CtrlN), was diluted in phosphate-buffered saline (without calcium and magnesium; Sigma-Aldrich, St. Louis, MO, USA) at 1:40, and immediately assessed as specified below.

The residual aliquots were diluted to a final concentration of $11 \times 10^6$ sperm/mL in a semen extender comprising Triladyl (Minitub GmbH, Tiefenbach, Germany), 20% (*w/v*) egg yolk, sugar, buffers, Tris, citric acid, glycerol, antibiotics, and distilled water. For the experimental groups, the extender was supplemented with 12.5 μM, 25 μM or 50 μM KAE (Sigma-Aldrich, St. Louis, MO, USA) in DMSO (dimethyl sulfoxide; Sigma-Aldrich, St. Louis, MO, USA), while the cryopreserved control group (CtrlC) was enriched with an equal amount of DMSO (final concentration of 0.5%). DMSO was used in CtrlC since it served as a medium through which KAE was delivered to the experimental groups. All diluted samples were loaded into 0.25 mL straws, cooled down to 4 °C for 2 h and subsequently frozen using a digital freezing machine (Digitcool 5300 ZB 250; IMV, Paris, France). Finally, the straws were plunged into liquid nitrogen and stored for one month. Before analysis, the straws were thawed in a 37 °C water bath for 90 s [7,10,18].

### 4.2. Sperm Motility

Sperm motility characteristics were assessed using computer aided sperm analysis (CASA; Version 14.0 TOX IVOS II.; Hamilton-Thorne Biosciences, Beverly, CA, USA) as previously published [10]. Ten microscopic fields were included in each analysis to ensure visualization of at least 300 cells. The analysis comprised the following parameters: motility (percentage of spermatozoa with a motility higher than 5 μm/s; %), progressive motility (percentage of spermatozoa with a motility higher than 20 μm/s; %), path velocity (μm/s),

progressive velocity (μm/s), track speed (μm/s), lateral amplitude (μm), beat frequency (Hz), straightness (%), and linearity (%).

### 4.3. Oxidative Profile

All control and experimental specimens were centrifuged ($300 \times g$, 20 °C, 10 min), washed with PBS twice and treated with a lysis solution composed of 150 mmol/L 1,4-dithiothreitol (Sigma-Aldrich, St. Louis, MO, USA), 25 mmol/L tris(2-carboxyethyl) phosphine (Sigma-Aldrich, St. Louis, MO, USA) and 2% β-mercaptoethanol (Sigma-Aldrich, St. Louis, MO, USA). The suspensions were vortexed for 5 min, diluted 1:1 in nuclease free water (Qiagen, Hilden, Germany), and then incubated with 200 μg/mL proteinase K (Sigma-Aldrich, St. Louis, MO, USA) at 56 °C for 2 h. DNA was extracted with the QIAamp DNA Mini Kit (Qiagen, Hilden, Germany), the yield and quality were determined using the GloMax®-combined spectro-flluoro-luminometer (Promega, Madison, WI, USA) at 260 nm [63]. The extent of oxidative damage to the sperm DNA was assessed with the EpiQuik™ 8-OHdG DNA Damage Quantification Direct Kit (EpiGentek Inc., Farmingdale, NY, USA) according to the instructions by the manufacturer [10]. The amount of 8-hydroxy-2′-deoxyguanosine (8-OHdG) was proportional to the OD intensity measured and is expressed in % [10,64].

Proteins from washed-out spermatozoa were extracted with RIPA buffer (Sigma-Aldrich, St. Louis, MO, USA) and protease inhibitor (Sigma-Aldrich, St. Louis, MO, USA). Following an overnight extraction, the samples were centrifuged at $11,828 \times g$ for 10 min at 4 °C and the supernatants collected for further analyses. Protein concentration was determined using the commercial Total protein kit (DiaSys, Holzheim, Germany) and the RX Monza instrument (Randox, Crumlin, UK) [65].

Oxidative damage to the proteins expressed through the levels of protein carbonyls was evaluated using the 2,4-dinitrophenylhydrazine (DNPH) method as previously published [10]. Protein carbonyls are expressed in nmol/mg protein.

Lipid peroxidation (LPO) expressed through the levels of malondialdehyde (MDA) was determined using the TBARS assay according to Tvrda et al. [18]. MDA concentration is expressed as μmol/g protein.

### 4.4. Western Blot

Randomly selected samples from each control and experimental group were treated with 4× Laemli buffer (BioRad, Hercules, CA, USA), β-mercaptoethanol and boiled at 95 °C for 10 min. The pre-treated samples were loaded (25 μg protein, 20 μL) into Mini-PROTEAN TGX Stain-free polyacrylamide gels (BioRad, Hercules, CA, USA), along with 7 μL of Precision Plus Protein marker (BioRad, Hercules, CA, USA). Gel electrophoresis was run for 2 h at 90 V; subsequently, the gel was visualized with the ChemiDoc Imaging System (BioRad, Hercules, CA, USA). The gels were transferred to PVDF membranes (Trans-Blot Turbo Pack; BioRad, Hercules, CA, USA) using the Trans-Blot Turbo Transfer System (BioRad, Hercules, CA, USA), at 7 min, 25 V and 2.5 A. The resulting membranes were incubated for $3 \times 10$ min in tris buffered saline (TBS), and then blocked either with 5% milk (Sigma-Aldrich, St. Louis, MO, USA; for BAX and Bcl-2) or 5% bovine serum albumin (Sigma-Aldrich, St. Louis, MO, USA; for HSP70 and HSP90) in TBS containing 0.1% Tween-20 (Sigma-Aldrich, St. Louis, MO, USA) at room temperature for 2 h. Subsequently, the membranes were exposed to the following primary antibodies against the proteins of interest overnight at 4 °C:

- Rabbit anti-BAX antibody (BCL2-Associated X Protein) N-Term, 1:1000 in 5% milk/TBS/ 0.1% Tween-20 (Antibodies Online; Dunwoody, GA, USA);
- Rabbit anti-Bcl-2 antibody (B-Cell CLL/lymphoma 2) N-Term, 1:1000 in 5% milk/TBS/ 0.1% Tween-20 (Antibodies Online; Dunwoody, GA, USA);
- Rabbit HSP70 Antibody, 1:1000 in 5% BSA/TBS/0.1% Tween-20 (Cell Signaling Technology; Danvers, MA, USA);

- Rabbit HSP90α (D1A7) mAb, 1:1000 in 5% BSA/TBS/0.1% Tween-20 (Cell Signaling Technology; Danvers, MA, USA).

The next day, the membranes were washed $5 \times 10$ min in 1% milk or 1% BSA, respectively, in TBS/0.2% Tween-20, and subsequently incubated with a secondary anti-rabbit antibody (GE Healthcare, Chicago, IL, USA) diluted 1: 15,000 in 1% milk or 1% BSA, respectively, in TBS/0.2% Tween-20 for 1 h. Finally, the membranes were washed $3 \times 10$ min in TBS/0.2% Tween-20 at room temperature. For protein visualization, the membranes were incubated with the ECL substrate (GE Healthcare, Chicago, IL, USA) for 5 min and processed with the ChemiDoc Imaging System. Relative quantification of the protein expression was performed with BioRad Image Software 6.1 (BioRad, Hercules, CA, USA). For all blots, the expression of a housekeeping protein was assessed. In this case, rabbit β-actin Antibody (Cell Signaling Technology; Danvers, MA, USA), diluted at 1:1000 in 5% BSA/TBS/0.1% Tween-20 was used [10]. The results are interpreted as relative quantification of the native control.

### 4.5. Statistics

Statistical analysis was performed with the GraphPad Prism program (version 9.2.0 for Mac; GraphPad Software, La Jolla, CA, USA). One-way ANOVA and Dunnett's test were selected for the analysis which was performed as follows:

- Native control (CtrlN) was compared to the cryopreserved control (CtrlC);
- Experimental groups were compared to both controls.

The level of significance was set at $p < 0.05$.

## 5. Conclusions

Summarizing the data collected in this study, we suggest that a dose of 25 μM kaempferol is the most suitable to improve the vitality of frozen–thawed bovine spermatozoa. Kaempferol has exhibited antioxidant properties that can prevent excessive damage to proteins, lipids, and DNA, crucial for the structural integrity and functional performance of spermatozoa. In addition, properly selected kaempferol doses can prevent the loss of proteins responsible to protect male gametes against cold shock whilst stabilizing the proportion of pro- and anti-apoptotic proteins in the cell. However, the specific involvement of kaempferol in specific intracellular and molecular signaling pathways must be further investigated in future studies.

**Supplementary Materials:** The following supporting information can be downloaded at: https://www.mdpi.com/article/10.3390/stresses3040047/s1, Figure S1: representative photograph of the loaded gel; Figure S2: Original HSP90 blot; Figure S3: Original HSP70 blot; Figure S4: Original BAX blot; Figure S5: Original Bcl-2 blot; Figure S6: Original Beta-actin blot.

**Author Contributions:** Conceptualization, E.T. and N.L.; methodology, Š.B., F.B., M.Ď. and E.T.; validation, E.T.; investigation, Š.B., F.B., M.Ď. and E.T.; resources, N.L. and E.T.; writing—original draft preparation, Š.B., F.B. and E.T.; writing—review and editing, Š.B. and E.T.; supervision, N.L. and E.T.; project administration, N.L. and E.T.; funding acquisition, N.L. and E.T. All authors have read and agreed to the published version of the manuscript.

**Funding:** This publication was supported by the Operational program Integrated Infrastructure within the project: Creation of nuclear herds of dairy cattle with a requirement for high health status through the use of genomic selection, innovative biotechnological methods, and optimal management of breeding, NUKLEUS 313011V387, cofinanced by the European Regional Development Fund, and by the Slovak Research and Development Agency grants no. APVV-15-0544 and APVV-21-0095.

**Data Availability Statement:** The data presented in this study are available upon reasonable request from the corresponding author.

**Acknowledgments:** We wish to thank the Center for Animal Reproduction (CeRA) Team of Excellence for their support.

**Conflicts of Interest:** The authors declare no conflict of interest. The funders had no role in the design of the study; in the collection, analyses, or interpretation of data; in the writing of the manuscript; or in the decision to publish the results.

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
