# Peer review of "Kaempferol Enhances Sperm Post-Thaw Survival by Its Cryoprotective and Antioxidant Behavior"

_stresses, doi:10.3390/stresses3040047_

Round 1

Reviewer 1 Report

Manuscript entitled “Kaempferol Enhances Sperm Post-Thaw Survival by Its Cryoprotective and Antioxidant Behavior”, submitted to Stresses, in my opinion contains many inconsistencies and inaccuracies. Manuscript requires major revision before publication.

I allowed myself to point out below my comments to the manuscript.

Abstract needs major corrections. The abbreviations of proteins BAX, Bcl-2 used in the abstract should be explained. Some sentences should be rewritten: line 14 directly related to cold storage”, line 16-17 (completely unclear), line 18 ensured” should be changed, line 22 whilst” should be deleted,

Please standardize the spelling of Bax/BAX in the whole manuscript.

Line 79 - KAE abbrevation should be used instead of kaempferol.

Line 84 – the” should be deleted.

Line 116 – MDA should be explained.

The HSPs abbreviation should be entered earlier - in line 127.

The abbreviation vs” should be write with a dot (vs.).

Abbreviations should be used consequently in the manuscript.

In the Results section 2.1. the Authors refered to motility, without specifying to which parameter of sperm motion. Please precize it.

Figure 1 remains unclear. Which parameter of sperm movement was considered here? Sperm motility” doesn't say much, because different sperm motility parameters are expressed as percentages. Does native state” means fresh spermatozoa (this is repeated in the captions of all figures)?

The Results section 2.2. (line 96) consists the 8-OHdG abbreviation which was not explained earlier in the manuscript.

Please correct the caption of Figure 8. Figure does not apply to HSP70.

Figure 9 – BCL-2 should be changed on Bcl-2.

Please explain the reason for adding DMSO to the control samples (lines 299-300).

The authors in the Material and methods section (4.2. Sperm Motility) did not specify which parameters of sperm motility were measured using the CASA system. Please add this information.

Manuscript entitled “Kaempferol Enhances Sperm Post-Thaw Survival by Its Cryoprotective and Antioxidant Behavior”, submitted to Stresses, in my opinion contains many inconsistencies and inaccuracies. Manuscript requires major revision before publication.

I allowed myself to point out below my comments to the manuscript.

Abstract needs major corrections. The abbreviations of proteins BAX, Bcl-2 used in the abstract should be explained. Some sentences should be rewritten: line 14 directly related to cold storage”, line 16-17 (completely unclear), line 18 ensured” should be changed, line 22 whilst” should be deleted,

Please standardize the spelling of Bax/BAX in the whole manuscript.

Line 79 - KAE abbrevation should be used instead of kaempferol.

Line 84 – the” should be deleted.

Line 116 – MDA should be explained.

The HSPs abbreviation should be entered earlier - in line 127.

The abbreviation vs” should be write with a dot (vs.).

Abbreviations should be used consequently in the manuscript.

In the Results section 2.1. the Authors refered to motility, without specifying to which parameter of sperm motion. Please precize it.

Figure 1 remains unclear. Which parameter of sperm movement was considered here? Sperm motility” doesn't say much, because different sperm motility parameters are expressed as percentages. Does native state” means fresh spermatozoa (this is repeated in the captions of all figures)?

The Results section 2.2. (line 96) consists the 8-OHdG abbreviation which was not explained earlier in the manuscript.

Please correct the caption of Figure 8. Figure does not apply to HSP70.

Figure 9 – BCL-2 should be changed on Bcl-2.

Please explain the reason for adding DMSO to the control samples (lines 299-300).

The authors in the Material and methods section (4.2. Sperm Motility) did not specify which parameters of sperm motility were measured using the CASA system. Please add this information.

Author Response

Dear reviewers, thank you for your questions. 

Please see the attachment file with the answers for your questions. 

Best regards, 

Authors of the manuscript. 

Reviewer 2 Report

Major comments and suggestions: 

1) All sperm motility characteristics (total, progressive, kinematic) must be included in the sections Results and Discussion.

2) Methodologically, what was the design of the study: sperm from each bull was evaluated separately? If yes, are the trends presented in Figures 1-4, and 6-8 valid for all bulls evaluated in the present study? Or the trends were different for some bulls, but Figures 1-4, and 6-8 present the average results? This must be explained very carefully in the appropriate section.

3) For WB, the pooled samples from all twenty bulls were used? Again, this must be explained in the appropriate section.

Minor comments:

a) What are the errors in the figures 1-4, and 6-9? SE? Please, clarify it in the figures' legends.

b) In figures 1-4, and 6-9 I recommend using letter symbols instead of asterisks to show the statistical differences on the level 0.05.

 c) Add details about the season of the year for semen collection.

I thank authors for their work.

I recommend minor editing of the English language.

Author Response

Dear rewievers, thank you for your questions.

Please see the attachment file with the answers.

Best regards, 

Authors of the manuscript.  

Round 2

Reviewer 1 Report

The document was improved according to Reviewer comments and in my opinion may be printed. 

The document was improved according to Reviewer comments and in my opinion may be printed.